# Study on the Types of Elderly Intelligent Health Management Technology and the Influencing Factors of Its Adoption

**DOI:** 10.3390/healthcare9111494

**Published:** 2021-11-02

**Authors:** Zhu Chen, Huiying Qi, Luman Wang

**Affiliations:** 1School of Nursing, Peking University, Beijing 100191, China; 2010108609@stu.pku.edu.cn; 2Department of Health Informatics and Management, School of Health Humanities, Peking University, Beijing 100191, China; wangluman@bjmu.edu.cn

**Keywords:** intelligent health management technology, the elderly, influencing factors

## Abstract

[Background]: In recent years, aging has become a global social problem. Intelligent health management technology (IHMT) provides solutions for the elderly to deal with various health risks. However, the elderly are facing many difficulties in using IHMT. Studying the application types of IHMT and the influencing factors of the elderly’s acceptance of it will help to improve the use behavior of the elderly. [Methods]: This paper summarizes the application types of IHMT, identifies the influencing factors of the elderly’s adaption of IHMT, and makes a systematic comment on the influencing factors. [Results]: We divide the different functions of IHMT for the elderly into four types: self-monitoring, medical care, remote monitoring, and health education. The influencing factors are divided into three types: individual, social, and technology. [Conclusions]: This study finds that IHMT’s application covers all aspects of the health services of the elderly. Among these applications, self-monitoring is the most used. We divided the influencing factors of the elderly’s acceptance of IHMT into three categories and nine subcategories, having 25 variables.

## 1. Introduction

In recent years, aging has become a global social problem, and has attracted extensive attention and research from scholars in China and abroad. The China National Health and Family Planning Commission estimates the Chinese population aged 60 and over reached about 255 million in 2020, accounting for about 17.8% of the total population [1]. The aging problem faced by China will be unprecedented in both speed and scale. Intelligent health management technology (IHMT) provides a series of solutions for the elderly to deal with the risks of physical and cognitive dysfunction, chronic diseases, the decline of social networks, and reduced physical flexibility. For example, in the health services of the elderly, significant investment has been made in mobile monitoring sensor networks, fall and loss monitors, and electronic health equipment.

IHMT in this study refers to the use of the Internet of things (IoT), cloud computing, and other information technologies to realize the health indicators of elderly users for real-time health monitoring and telemedicine services through the Internet, intelligent sensing devices, and other means. To date, many scholars have undertaken further research on the use of IHMT in the elderly. For example, by recruiting Chinese Canadians over the age of 65 with hypertension, a previous study used a smartphone application (app) to carry out healthy diet education and control hypertension, to verify the availability and feasibility of the app [2]. The rise and popularity of wearable devices have also promoted relevant research. Wearable devices have been used for elderly patients with glycosylated hemoglobin to realize real-time automatic collection and storage of blood pressure data [3]. Telemedicine also provides convenient medical services for the elderly at home [4]. Electronic health record systems and patient portals play a vital role in supporting the self-health management of the elderly, improving the efficiency of medical services, and reducing medical costs [5].

Chung et al. concluded that when smart technology is implemented appropriately and ethically, it can strengthen older adults’ quality of life, safety, and prospects for aging-in-place [6]. Nevertheless, those over 60 are generally slow to accept information technology, and it is often difficult for the elderly to use this technology. Thus, what forms of IHMT can be applied to the elderly? What factors affect the elderly’s acceptance of IHMT? At present, there are few studies on the above problems. This paper systematically reviews the relevant research on the use of IHMT by the elderly, generalizes the types of IHMT, and identifies the influencing factors of the elderly’s acceptance of IHMT to provide a valuable reference for subsequent study.

## 2. Materials and Methods

Most of the vital relevant research is published in core international journals. Therefore, the Web of Science core database was used as the data source in this paper. The search strategy was as follows: Topic= (① “old” or “senior” or “elderly”; ② “health management”; ③ “technology” or “eHealth” or “telemedicine” or “telemonitoring” or “telecare” or “assistive technology”; ④ “use” or “adoption” or “adherence” or “effect” or “impact”). During the retrieval process, we found that the relevant literature was mainly published after 2011, so we limited the publications to the articles published since 2011 as the data source of this study. The retrieval time was up to 7 June 2021.

The inclusion criteria were as follows: ① English articles with literature type of “article”; ② the main subjects of the study were the elderly aged 60 and over; ③ the theme of the study was the influencing factors and effects of IHMT used by the elderly. First, we searched a total of 961 papers. By browsing the title, abstract, and full text, in turn, those papers that met the research theme of this paper were selected. The selection process was conducted using the Preferred Reporting Items for Systematic Reviews and Meta-Analyses extension for Scoping Reviews (PRISMA-ScR) [7] and is summarized in Figure 1. Specifically, we first deleted four duplicates, and then removed three non-English papers. Through title screening, we deleted 565 articles unrelated to the elderly and information technology. We read the abstract and removed 185 articles unrelated to adopting and using information technology for the elderly and health management. We further excluded 101 papers not related to the research topic through full-text reading. We separated 31 review articles, and 72 articles highly related to the topic, which finally comprised the study’s data source.

## 3. Results

### 3.1. Types of IHMT

According to the different functions it provides, we divided IHMT into four types: self-monitoring, medical care, remote monitoring, and health education. Table 1 lists the specific technologies adopted by each of these types. We should note here that IHMT often uses a software system formed by mixing multiple technical types of equipment. Nonetheless, whether it comprises a single specialized device or numerous types of technical equipment, it comprises an application to provide services for the elderly. Only the primary technologies are listed here.

#### 3.1.1. Self-Monitoring

Self-monitoring is a technology used to provide the elderly with the measurements and early warnings of various physical health indicators, and enable elderly to be at home safely. The authors—Reeder, McManus, Quinn, Willard, Hermanns, et al. showed that self-monitoring can be realized in the form of medication-dispensing devices, embedded sensors, self-management applications, wearable systems, online community care platforms, and physical activity trackers [8,9,11,15,16,17]. In their study, Reeder et al. found that home-based devices for medication management and health monitoring can automatically distribute pre-installed drugs and send an alarm to individuals about the medication time. If the drugs are lost or the machine needs to be refilled, the telemedicine staff are notified [8]. McManus et al. found that the smartphone app is one of the primary forms of mobile health tools widely used in vital signs monitoring. A smartphone app using embedded sensors can monitor and manage the patient’s heart rate, automated discriminate arrhythmia, and atrial fibrillation. Thus, atrial fibrillation screening can be carried out [9]. Portz et al. focus on another heart failure management app that can regularly monitor and track the heart failure indicators of patients to provide solutions for the elderly to self-manage the symptoms of heart failure [10]. Because there are now an increasing number of elderly patients with diabetes, the use of self-management apps for managing diabetes is of great significance. Based on blood glucose values, Quinn et al. found that an app can provide patients with automatic messages and educational information conducive to managing their symptoms themselves [11]. For managing other chronic diseases, such as hypertension and mental health problems, corresponding self-management app provide intervention for specific patients. Related research was mentioned in the study of Still, Gould et al. [13,14], which examined an online community care platform designed for the elderly living at home. This allows frail elderly to be independent and supports their functioning by stimulating self-care and providing reliable information, products, and services, which favor managing and arranging their care and support as much as possible [15]. For the elderly with Parkinson’s disease, Hermanns et al. demonstrated that a physical activity tracker can assess their physical activity and, as a result, increase their physical activity and consequently improve their quality of life (QOL) [16]. The study of Batsis, Zaslavsky, and Zhang et al. confirmed that smart wearable devices, as cutting-edge technology, play an essential role in self-monitoring technology used in monitoring individual indicators. Specifically, these devices can record data to classify an individual’s activity level [17], record sleep data [18], and monitor blood pressure using a wearable blood pressure monitor [19], and ultimately improve the health level of the user.

#### 3.1.2. Medical Care

Medical care is the technology that provides support and services for the medical treatment, daily health care, and physical health of the elderly. In the study of Chen, Parker, Taha, Kullgren, McMahon, Portz, et al.’s, medical care mainly provides telemedicine, apps, electronic health record systems, wearable systems, physical activity monitors, and patient portals [21,23,28,31,32,33]. Telemedicine is the remote diagnosis and treatment of the elderly using telecommunications technology. To manage chronic diseases [21], improve medical quality and efficiency, and control costs [22], Chen, Zhou et al. showed that telemedicine provided a range of telehealth services remotely through various telecommunication tools, including landline phones, mobile phones, smartphones, and other mobile wireless devices. The mobile app is one of the most crucial service forms of medical care technology. Parker et al. found that a cerebral palsy pain management app program helped the elderly manage the pain by monitoring treatment side effects and pain degree [23]. Chung, Goransson, Lee, et al. conducted following similar research, examining an app that supports cognitive behavioral therapy for insomnia [24]. Another app helps the elderly receive home care [25], and a health management app uses mobile health technology and health games [26]. Thus, apps can provided different services in different aspects of daily health care and physical health of the elderly.

In addition to being used by healthcare providers, electronic health record systems provide personalized services for the elderly [27,28,30]. Wearable systems record and analyze the vital data of the elderly by monitoring their critical vital signs and exercise status, and provide adequate support for their medical health through remote nursing plans and remote consultation. For example, in the study of Kullgren et al., to encourage the elderly to walk more, a wearable pedometer was applied. A daily goal was set to enable them to complete the desired number of steps, encourage them to participate in regular exercise and adopt good habits, and, as a result, promote their health [31]. McMahon et al. showed that physical activity monitors have the potential to facilitate tracking practices [32].

Portz et al. showed that the patient portal provided patients with personal health information, medication, and examination information to participate in their health management [33]. In terms of the web-based system, a remote medication adherence measurement system developed by Brath et al. realized the measurement and management of patient drug compliance, and was accepted by patients [34].

#### 3.1.3. Remote Monitoring

Remote monitoring is a technology that enables monitoring of the elderly outside of conventional clinical settings (e.g., in the home), increasing access to care and decreasing healthcare delivery costs. The authors—Turchioe, Chau, Sheeran, Taha, Creber, Bakas, et al. found that the most common are telemonitoring apps, web-based systems, in-home monitors, patient portals, embedded sensors, and robotics [4,35,36,38,39,40]. In their study, Turchioe et al. proved that measurement information systems for patient-reported outcomes can monitor the elderly with chronic illnesses outside of healthcare settings. Data of the elderly could be transmitted through the symptom monitoring app [35]. Chau et al. showed that the web-based system can send the physiological indexes of the elderly to the network platform [4]. Gellis et al. confirmed that installing a small, tabletop in-home monitor in the elderly’s home can effectively be used to monitor heart or chronic respiratory failure [37]. In addition, by allowing patients to perform health management tasks, Taha et al. found that the patient portal can also transform healthcare by providing patients with increased access to personal health information [38]. Creber et al. found that the embedded sensor can provide regular symptom monitoring for elderly patients with heart failure to improve prevention and care, and enhance the QOL for the elderly [39]. Furthermore, robotics is a relatively high-end monitoring technology. In terms of app, Bakas et al. showed that telepresence robot technology can provide two-way video-mediated communication with remote in-home navigation for the elderly living independently in the retirement community to optimize healthy living [40].

#### 3.1.4. Health Education

Planned, organized, and systematic health education activities consciously promote people to adopt healthy behaviors and lifestyles. Health education can enhance the awareness of self-management of elderly users, improve treatment compliance, and reduce disease risk. Health education technology can help the elderly understand which behaviors affect their health, and to reduce or eliminate the risk factors. Studies by Zou, Richard, et al. demonstrate that an app and a web-based system can carry out health education [2,41]. In their study, Zou et al. applied a diet education app that provided proper diet knowledge for elderly users, supported them in forming a healthy diet, and controlling hypertension [2]. Richard et al. proved that a web-based system can become an educational tool, covering many people using different online platforms. For example, a web-based health consultation system provides personalized suggestions to the elderly through a coach-supported interactive Internet platform, improving the cardiovascular risk profile and reducing the risk of cardiovascular disease and cognitive decline [41]. Another example studied by Vanoh et al. is a web-based educational tool called WESIHAT 2.0, which can systematically educate older adults about precautionary strategies against mild cognitive impairment (MCI) and reduce their risk of MCI [42].

### 3.2. Influencing Factors of the Elderly Adopting IHMT

A recent survey by American Association of Retired Persons (AARP) on technology trends among older adults identified smart technology as an emerging market for the 50+ age group, with many adults interested in buying smart technology products, but only about 10% currently using them [43]. The adoption of IHMT by the elderly is affected by many factors, as shown in Figure 2. As the main body of use, whether to adopt intelligent health management technology is influenced by objective factors (age, education, information literacy, etc.) and subjective factors (health awareness). Society is composed of people’s social structure, social customs, values, and other content. It affects people’s lifestyle (such as social attributes) and consumption concepts (such as source reliability, service, and cost), and directly impacts whether the elderly adopt intelligent health management technology. IHMT provides support and guarantees for the health management of the elderly. Its usefulness, ease of use, and safety will affect the user experience of the elderly. Therefore, this paper reviews the influencing factors of the elderly’s adoption of IHMT from the three levels of individual, society, and technology.

#### 3.2.1. Individual Level

From the individual level, the factors affecting the adoption of IHMT by the elderly are mainly reflected in two aspects: objective factors and subjective factors, as shown in Table 2.

Objective factors

The authors—Rasche, Taha, Batsis, Irizarry, Stillet, McManus, et al. proved that the elderly’s adoption of IHMT was affected by age, education level, region, e-health literacy, information literacy, ethnicity and language, and physical difficulties [5,9,12,13,28,48]. First, the user behavior depends on age to some extent. Gordon et al. showed that the number of smartphones decreases with age, and the convenience of using these devices also decreases with age [44]. In Taha et al.’s study, the older the age, the worse the ability to accept IHMT [28]. It is commonly found that most IHMTs are based on modern communication technology and network systems, which is a significant obstacle to promoting IHMT among the elderly.

Regarding the education level, the higher the education level, the greater the inclination to use new technologies. Nguyen et al. showed that high health literacy indicated that people had enough health knowledge to understand and manage their health status [49]. Electronic health literacy represents the ability of the elderly to use information technology to solve health problems. Irizarry et al. found that the elderly with a high level of electronic health literacy had a higher acceptance of new technologies [5]. Similarly, Taha et al. proved that people skilled in computers and the Internet have high information literacy, which means they have better Internet skills and computing skills, and more vital application ability in new technologies. These people are more inclined to adopt IHMT [38]. In Wang et al.’s study of the community-dwelling older adults in Hong Kong, it was found that the eHealth app for self-management is less likely to be used by older, unmarried, less educated, unemployed, and lower-income cancer survivors [56].

From the region’s perspective, those in rural areas are less likely to use new technologies than those in cities. Batsis et al. found that there are many causes for this phenomenon; one main reason is that, due to geographical constraints, they have much less access to new technologies [48]. Gordon et al. proved that there are also significant racial/ethnic differences in eHealth technology among the elderly. Specifically, the prevalence of eHealth technology among blacks, Latinos, Filipinos, and the Chinese elderly is lower than that of whites. Whites and Chinese older people are more likely to become Internet users than black, Latino, or Filipino older people [44]. In McManus et al.’s study, the physical difficulties resulting from disease and aging for the elderly are also one of the hindrances to accepting and using new technologies [9].

2.Subjective factors

Subjective factors are the internal driving force that causes people to engage in a specific behavior. The studies of Reeder, Quinn, Zhang, Rasche, et al. found that demand, self-efficacy, compliance, and skeptical attitudes are subjective factors that affect the adoption of IHMT by the elderly [8,11,12,19]. Self-efficacy refers to the notion that one can successfully execute the behavior required to produce the outcome; that is, a person believes that he can be successful when carrying out a particular task. Choi et al. showed that self-efficacy can enable the elderly to have a high level of health self-efficacy and have better health-promoting behaviors [30]. The study by Middlemass et al. found that the initial installation process of remote monitoring equipment is critical to improving self-efficacy, and can promote their study [50]. Nguyen et al. proved that the elderly who have high confidence in managing their situation and are willing to carry out self-care have higher self-efficacy and a more vital willingness to accept and use IHMT [49]. The studies of Nymberg, Askari, Hermanns, et al. found that those who expressed “distrust of their abilities” had lower self-efficacy [45]. This idea then becomes an obstacle to adopting new technologies and harms their medical applications [51]. Thus, it is vital to continue encouraging self-efficacy [16] to more actively engage elderly in health promotion behavior.

In addition, as an essential index to evaluate the effect of IHMT, compliance can subjectively improve the health of the elderly. For example, Chung et al., summarized that cognitive behavioral therapy for insomnia app is usable among older users and can improve subjective sleep quality after a 1 week self-help intervention period [24]. Furthermore, Mira et al.’s study showed that a medication self-management app (called Alice) improves patients’ compliance, helps to reduce the forgetting rate and medication error rate, and improves the perceived independence of drug management [52]. Mansson et al.’s study found that, in self-administered interventions for fall prevention, the elderly who adopted the corresponding technology showed higher satisfaction than those who do not use technology. Compliance appeared to facilitate regular exercise after the intervention period [55]. The improvement of compliance reflects the utility and potential of this technology in terms of application. It is more likely to become an excellent auxiliary for user health management in the future. The skeptical attitude is a negative factor affecting the adoption of IHMT by the elderly. The obstacles found in Rasche et al.’s study were lack of trust, data privacy, and fear of misdiagnosis [12]. Perceived demands reflect an individual’s concern for their health. Reeder et al. found that individual perceived needs directly impact the use of new technologies [8]. In Chen et al.’s study, sensing the threat of illnesses and the severity of their diseases are reasons for them to accept and use new technologies; that is, the emergence of disease threats can stimulate their willingness to maintain or restore health [21]. Wang et al. proved that the positive attitude towards IHMT also promotes the demand of the elderly for new technologies [56]. 

#### 3.2.2. Social Level

With the exception of personal factors, users are not isolated but are also affected by the social environment and other users. Therefore, at the social level, we review the influencing factors of the elderly’s acceptance and use of IHMT from four aspects: social attributes, source credibility, service, and cost, as shown in Table 3.

Social attributes

Social functioning is a positive for the adoption of IHMT as this broadens their applications and improves their value. The study by Huh et al. found that users showed a stronger preference for the social module. The elderly showed a favorable response on sharing wellness information with their health care providers, which they saw as a potential means of increased patient-doctor communication [57]. Research in information sharing showed that the elderly want health care providers to monitor them, and they are interested in small social networks and intimate relationships [58]. Supplementing cognitive training systems with social, emotional, or recreational functions may improve adherence to their use [60].

2.Source credibility

Considering that all types of IHMT provide users with many forms of health information, advantages and disadvantages coexist. In addition, the elderly cannot clearly identify detailed information, so equivocal or unscientific information will harm their health. Thus, source credibility becomes an influential critical factor in the adoption of new technologies by the elderly. Health professionals (doctors, pharmacists, nurses) are an essential source of people’s access to medical health information, so the recommendation of professionals is also one of the influencing factors for users to accept or use IHMT. Research has found that health providers’ support helps heighten system/device credibility. Their affirmation gives the system/device some authority and reliability, which also leads to acceptance of their quality. As mentioned in Middlemass et al.’s study, one patient felt that his general practitioner had personally recommended him for the study (rather than being identified from a computer search) and found that very reassuring [50]. In addition, Wang et al. proved that family’s/friends’ opinions can also significantly affect the elderly acceptance of IHMT [56].

3.Service

Excellent services can promote the use of IHMT by elderly, including the encouragement and training of medical staff. The reason for the high usage of a home care app may have been the trust the elderly had in the homecare nurses when asked to participate in the study. The elderly describe that trust, which makes them feel modern and acknowledged as valued people. In conclusion, the home care nurses may have played a role in inspiring the elderly to use the app [25]. Irizarry et al. proved that both high and low health literacy groups felt that specific task-based training was necessary [5]. Even people with high health literacy need some training to better master technology and use equipment. In addition, if the users’ training willingness can be clarified and their training needs met, the existing digital gap can be addressed, which will be conducive to the promotion of IHMT [13]. In terms of service implementation, collaboration with the community to provide workshops, information sessions, support groups, and social media interviews will promote the use of related apps in the community [2]. Navigation functions can assist in supporting obesity wellness intervention with the advantage of enhancing health promotion in a remote, geographically constrained community [48]. By comparison, Sun et al. found that elderly patients needed sufficient time to become familiar with the mobile medical system, which required a certain amount of time and frequency for training. Then, they gradually mastered and skillfully used it through repeated operation [62].

4.Cost

Whether to save operating costs and apply limited medical resources to improve the efficiency of clinical medical services are important considerations for health care institutions. Those technologies that perform better in saving human, material, and financial resources are more popular with healthcare institutions. Kastner et al.’s study conducted a cost description analysis of an app related to chronic disease, evaluated the cost of the overall and staged implementation, and generally assessed the technology’s application [63].

#### 3.2.3. Technical Level

From the perspective of technical realization, only products that are attractive to the elderly and can well meet their needs can make them willing to accept and use the technology. We analyzed the influencing factors at the technical level in terms of three aspects: perceived usefulness, perceived ease of use, and security, as shown in Table 4.

Perceived usefulness

Perceived usefulness is mainly expressed in helping elderly users achieve health management according to the specific situation of individuals. The authors—of Kullgren, Zhou, Turchioe, Irizarry, et al. showed that perceived usefulness is reflected in functionality, information quality, personalization, and interactivity [5,22,31,35].

Whether the elderly are interested in intelligent health management is related to its core functions (such as reminders, notifications, incentives, tracking, and goal setting) and how it is provided. For example, as a function preferred by the elderly, reminders are the most convenient function. Nevertheless, the time and frequency of reminders need to be well designed, or the elderly will only ignore them. For instance, in the study by Kuo et al. of taking medication and confirming the dosage, the researchers announced each prescription via the push notification service, and the application finally prompted the elderly to take their medicine [65]. Motivating mechanisms, such as comparison with other users’ behavioral data, are also considered to promote users’ usage. Most users like tracking functions that are thought to increase health awareness and can observe health behavior progress. The goal-setting function is also welcomed by the elderly, who feel that it helps self-regulation and gradually changes their behavior. Moreover, the goal-setting function is better able to coordinate tracking, real-time feedback, and progress reporting. For example, Kullgren et al. found that pedometers, goal setting, and regular feedback on goal achievement were as effective as economic incentives, peer networks, or a large number of interventions with both [31].

In addition, the information quality provided by IHMT also affects the elderly experience [65]. When poor-quality information repeatedly appears, the elderly are more likely to give up using it. Information quality has a significant effect on the acceptance of telemedicine by the elderly, as shown in the research of Zhou, M. et al. [22].

For the elderly, personalization has also become a key feature to increase attractiveness and acceptability. The Patient-Reported Outcomes Measurement Information System (PROMIS), an inclusively designed mobile application in Turchioe et al.’s study, can monitor patients with chronic illnesses. It integrates biological, psychological, social, and medical research in the support of independent living [35]. The study by Kim, H. et al. also showed that personalized data and timely response seem to enhance the participation of the elderly in mobile medicine [70]. Personalized applications can identify when learning may produce the best results, such as accepting a new prescription, solving patient problems, and supporting problem-based learning [71]. Hence, the tailor-made characteristic of an intervention that is specifically suited to the needs of older individuals fits with the current development towards a more personalized approach in medicine.

The core functions allow users to accept and use technology, and interactivity offers the emotional support needed to maintain motivation. As a promising method of providing health education and services, health games are also used to enhance self-care behavior and increase treatment compliance [26]. The interactive application creates new interaction between family and nurses [27], and provides a good communication platform between patients and health professionals. However, it is worth noting that this web-based online interaction cannot replace direct clinician–patient interaction, which is necessary to clarify any potential issues [5]. In high-level self-monitoring applications, gamification is more stimulating and pleasant, thus enhancing users’ optimism and enthusiasm. The elderly hope to “get in touch with doctors no matter how they contact,” which is very important [45]. Most users prefer to use interactive applications rather than one-way information delivery systems [70]. Regarding highly interactive health management systems, successful integration of the system into daily routines and ongoing interaction was found to lead to positive aspects of self-perception and embodiment, and ultimately supported the long-term health management of the elderly [72].

2.Perceived ease of use

Perceived ease of use refers to the labor-saving degree of service, which significantly determines whether the elderly accept IHMT. Many studies have shown that perceived ease of use is positively correlated with the acceptance and use intention of the elderly [42,51,73].

The ease of use is mainly reflected in the user interface design and efficiency, which is particularly important for the elderly [74]. Vanoh et al.’s study argued that a friendly program interface can considerably improve the acceptance of the elderly. However, when the text on the interface is too small, it is difficult for the elderly to interact with the application; when the displayed text’s readability is poor, the elderly cannot be encouraged to read with greater interest [42]. Conversely, a clean and simple interface can help the elderly navigate the application [46]. For the elderly with specific needs, a unique design is required to meet the requirements. In the case of older people with poor vision, macular degeneration, or glaucoma, or those with decreased auditory function, customized and simplified technologies are necessary, such as expanding the screen and enhancing volume control, which makes the technology easy for them to use [23].

In terms of efficiency, the application of touch screens [42] and voice input [75] can facilitate the use by the elderly and improve the use efficiency.

3.Security

Health data is always associated with security and privacy issues. Users’ concerns about health data security and privacy are one of the influences on the adoption of IHMT for many elderly people. Technology that does not consider these aspects will raise concerns about security and privacy issues [14]. A lack of clarity about how IHMT handles personal privacy data will lead to users’ anxiety [51].

The reliability of technology is also one of the concerns of the elderly. Concerns about misdiagnosis by programs or equipment due to unreliability is one of the main obstacles affecting IHMT for the elderly [12]. The research of Middlemass, J. B. et al., showed that unreliable technology is linked to poor Internet connectivity and data transmission in rural areas, and led to the creation of several technical alerts. These circumstances could adversely affect patients’ perception of reliability [50].

## 4. Discussion

Regarding the types of IHMT, it is observed that the application of IHMT covers all aspects of elderly health services. Among these applications, self-monitoring is the most used in monitoring individual indicators and managing chronic diseases. Mobile apps are the most widely used from the perspective of specific implementation technology. The main reason for this is that smartphones integrate fine sensor technology and Internet technology. They can complete many health management and monitoring tasks without other intelligent devices. In addition to the high popularity of smartphones, combining various types of IHMT with smartphones in the form of mobile apps can provide convenient services for the elderly. Moreover, various technology combinations often constitute intelligent systems, such as wearable blood pressure monitoring systems, including wristband devices and mobile phone app programs. A wristband has been used for ambulatory blood pressure monitoring, and an app program used to record and transmit data [19]. An automatic collection and storage sphygmomanometer was combined with a network to form an intelligent wearable system [3]. Smart health technology was combined in a novel acoustic monitoring technology and an online learning recommendation system to identify mood states and provide mindfulness-based stress management tools [66].

The summary of the objective factors affecting the adoption of IHMT by the elderly showed differences in age, living area, education level, e-health literacy, and physical health status. Many studies have been conducted on the impact of age, e-health literacy, and information literacy. Older age, lower e-health literacy, and information literacy are all negative factors in the adoption of new technologies by the elderly. Thus, a remaining question relates to how to improve the electronic health literacy and information literacy of the elderly and lower the barrier of entry of IHMT so that the elderly can easily use these devices. Improving electronic health literacy requires joint efforts from two aspects. One is to strengthen the health education and information literacy of the elderly group to ensure they remain aware of and familiar with current technologies. In addition, the technical design should provide a more convenient and easy-to-use design according to the use characteristics of the elderly. By summarizing the subjective factors, we found that the perceived needs, self-efficacy, compliance, and skeptical attitude of the elderly will affect their adoption and use of IHMT. Further studies have undertaken on self-efficacy and adherence. These two indicators are often used to describe the use effect of specific technologies over a long period. It was concluded from many studies that a good level of self-efficacy and high compliance can stimulate the internal long-term use intention of the elderly, thus making it easier for the elderly to use specific technology for a long time.

Numerous studies have been conducted on the influencing factors of IHMT adoption by the elderly in terms of social attributes and services from the social dimension. The social function provided by IHMT can increase the interest and sense of achievement in the use process, and thus improve users’ compliance. The elderly also want technical training so that they can better master their use of IHMT. These studies indicate the solid social needs of the elderly as a vast social group, while also showing that the elderly want to obtain social services. Therefore, paying attention to the needs of the elderly in the social dimension will be a future expansion direction of IHMT.

The analysis of the factors affecting the elderly’s adoption of IHMT at the technical level shows that perceived usefulness and ease of use are usually used together as indicators to measure the elderly’s acceptance of technology. Successful research results have been obtained relating to functionality, personalization, and interaction in perceived usefulness. Regarding perceived ease of use, many studies have related to use efficiency. Only proper, easy to use, safe, and reliable products, designed for the elderly, are welcomed by the elderly. Consequently, the development and improvement of IHMT in the future should improve perceived usefulness and perceived ease of use, ensure data security, and maintain a balance between these factors.

## 5. Limitations

The literature search scope of this paper only covers English-language research retrieved from the Web of Science core database. Conferences, books, and doctoral theses were not within the scope of retrieval. Hence, there are certain limitations in the coverage of the literature, and the conclusions will also have certain restrictions.

## 6. Conclusions

This paper systematically reviews the journal literature relating to the use effect of IHMT for the elderly. Four types of IHMT for the elderly are summarized. The influencing factors of the elderly’s acceptance of IHMT are divided into three categories and nine subcategories, having 25 variables. It is hoped that the results of this research can provide a reference and enlightenment for the promotion and application of IHMT for the elderly.

## Figures and Tables

**Figure 1 healthcare-09-01494-f001:**
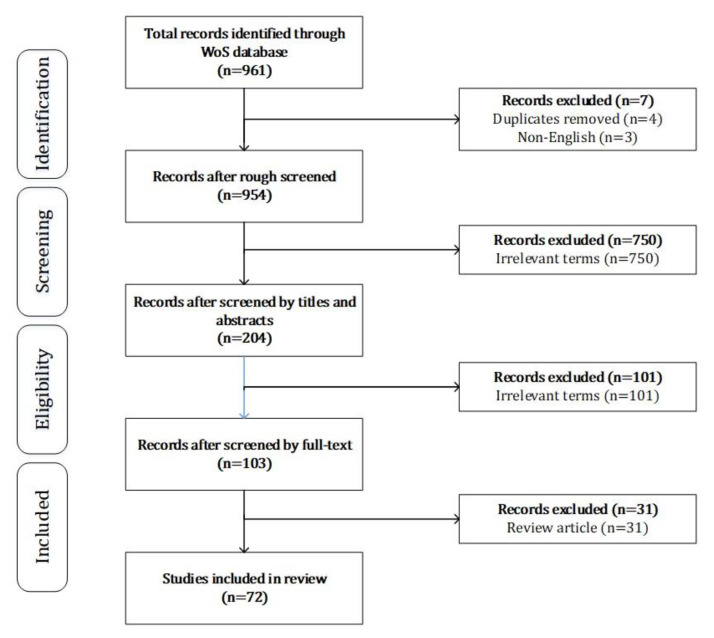
Article search and screening process.

**Figure 2 healthcare-09-01494-f002:**
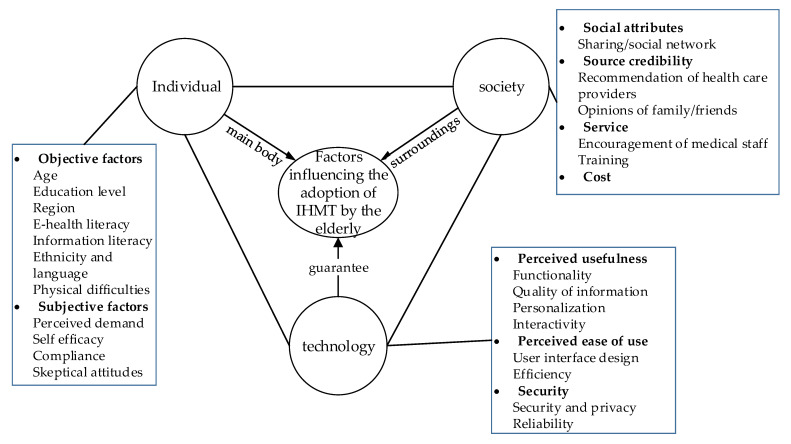
Relationship chart of influencing factors of the elderly adopting intelligent health management technology.

**Table 1 healthcare-09-01494-t001:** Common technologies of intelligent health management.

Types (Number of Related Papers)	Specific Techniques	Representative Literature
Self-monitoring (34)	Medication dispensing device	[8]
	Embedded sensor	[9,10]
	Self-management application	[11,12,13,14]
	Online community care platform	[15]
	Physical activity tracker	[16]
	Wearable systems	[3,17,18,19]
	Mobile app	[20]
Medical care (21)	Telemedicine	[21,22]
	Mobile app	[23,24,25,26,27]
	Electronic health record systems	[28,29,30]
	Wearable systems	[31]
	Physical activity monitors	[32]
	Patient Portal	[5,33]
	Web-based system	[34]
Remote monitoring (12)	Telemonitoring app	[35]
	Web-based system	[4]
	In-home monitor	[36,37]
	Patient Portal	[38]
	Embedded sensor	[39]
	Robotics	[40]
Health education (5)	Mobile app	[2]
	Web-based system	[41,42]

**Table 2 healthcare-09-01494-t002:** Influencing factors at the individual level.

Influencing Factor	Related Variable	Main Conclusion	Representative Literature
Objective factors	Age	The older the elderly, the worse their performance	[12,28,44,45,46]
	Education level	Populations with a high education level tend to use new technology	[28,47]
	Region	Compared to urban residents, rural ones are not likely to use new technology	[48]
	E-health literacy	E-health literacy is considered as a prerequisite of using mobile health app	[5,15,30,47,49]
	Information literacy	People skilled in computer and Internet use tend to adopt IHMT	[13,14,38,46]
	Ethnicity and language	Ethnic and linguistic differences affect users’ acceptance and use behavior	[13,44]
	Physical difficulties	Physical difficulties are an obstacle for the elderly to accept new technologies	[9]
	Self-efficacy	People with high efficacy are more willing to accept new technology	[11,13,16,30,39,45,49,50,51]
	Compliance	Procedures/equipment that can improve compliance are easy to be continuously used	[19,24,29,52,53,54,55]
	Skeptical attitudes	Skepticism is a barrier to the elderly using new technology	[12]
	Perceived demand	Perceived demand is an essential factor in promoting the use of intelligent health management devices	[8,21,56]

**Table 3 healthcare-09-01494-t003:** Influencing factors at the society level.

Influencing Factor	Related Variable	Main Conclusion	Representative Literature
Social attributes	Sharing/social network	Social networking increases interest and a sense of achievement and promotes user’s compliance	[57,58,59,60]
Source credibility	Recommendation of health care providers	The support of health care providers helps to improve the credibility of the system/equipment	[50,61]
	Opinions of family/friends	The opinions of family/friends affect users’ acceptance of health information technology	[56]
Service	Encouragement of medical staff	The encouragement of medical staff has played an important role in the use of health information technology by the elderly	[25]
	Training	The elderly look forward to technical support through training	[2,5,13,14,39,48,60,62]
Cost	Cost	Whether to save operating costs is a central consideration for health care organizations	[10,21,37,63]

**Table 4 healthcare-09-01494-t004:** Influencing factors at the technical level.

Influencing Factor	Related Variable	Main Conclusion	Representative Literature
Perceived usefulness	Functionality	Users are interested in core functions such as speed of running, reminders, notifications, encouragement, follow-up, goal setting, and online video presentation, and in the way they are provided	[31,34,64,65,66]
	Quality of information	Information quality has a significant impact on older people’s acceptance of intelligent health management technologies	[22,67]
	Personalization	Personalization is a key characteristic that enhances the attractiveness and acceptability of IHMT	[35,41,68,69,70,71]
	Interactivity	Interactive games and healthcare worker or peer interactions can increase interest in older adults	[5,17,26,27,45,58,62,70,71,72]
Perceived ease of use	User interface design	Devices with simple operation, large font, and touch screen interface are more popular	[23,26,42,46,48,68,73,74]
	Efficiency	Voice input design can improve time efficiency	[75]
Security	Security and privacy	Users’ concern about the security and privacy of health data is one of the reasons they do not use or do not continuously use.	[14,51]
	Reliability	Fear of misdiagnosis due to unreliability of programs/equipment is one of the main obstacles	[12,50]

## Data Availability

Not applicable.

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
