# Peer review of "Study on the Types of Elderly Intelligent Health Management Technology and the Influencing Factors of Its Adoption"

_healthcare, 2021, doi:10.3390/healthcare9111494_

Round 1
Reviewer 1 Report
To my knowledge, this is the first study to give a broad overview of the use of 'Intelligent Health Management Technology', as the authors call it, by the elderly, with pitfalls and possible solutions. In doing this, the authors need to be congratulated with undertaking this effort. The authors collected information from a large number of publications on the topic, and used this information for this overview.
However, in its present form the manuscript is hardly readable due to suboptimal English, with many strange and incomprehensible sentences. I would recommend to have the manuscript completely revised by someone fluent in English.
Secondly, some comments on the content:
- In general, I have the feeling that the subject has been divided in to many sub-items and sub-sub-items. For instance, there is much overlap in the text in subchapters 3.1.1, 3.1.2 and 3.1.3. Also, I find the distinction between 3.2.1, 3.2.2 and 3.2.3 (individual, social, technical) confusing. The authors may try to make the distinction clearer to the readers.
- The setup of the manuscript is that statements or conclusions from referenced articles are given in the text. However, the way this is done is often not in the 'normal way'. To give an example: on page 8, second sentence under 1. Social attributes: "The study found that users .... [54]". The correct way to write this is: "The study by Huh et al found that users ... [54]". Another example on page 7: "For example, the cognitice behavioral ... [23]. For one, this last sentence is completely incomprehensible, and secondly, it seems like the authors have studied this themselves, only after seeing the [23] one can conclude that this conclusion comes from a referenced article. Please correct all this where applicable.
- Secondly on this topic of all conclusions based on referenced articles: often in the text the referenced article is just barely described, which almost brings the need for me as reviewer, but also for the reader, to obtain and read each of the referenced articles to understand what the authors (of the present manuscript) intend to prove.
Then a number of short comments / examples of strange or even incomprehensible wordings (these are just a few examples):
- last sentence of first paragraph: "health equipment have been heavily .." -> incorrect English
- 2nd paragprah, 2nd sentence: "At present.. elderly IHMT" -> what is "elderly IHMT"?
- page 2, "Bradford's law": I did not know what Bradford's law was. I looked it up, but feel that it is irrelevant here. Just leave out this sentence (on Bradford's law), certainly without further explanation.
- subchapter 3.1.4, in the middle of the paragraph: "Web-based system can become a network-based educational tool by using different online platforms" -> Completely incomprehensible what is meant here.
- page 7: What exactly is meant with "self-efficacy"?
- page 8: "..low self-identification ability of the elderly" -> What is that? The whole sentence, by the way, is incomprehensible.
- page 9: "Besides, if we can well meet the training willingness ..." -> Bad English, and possibly for that reason, totally incomprehensible.
The term IoT is mentioned a number of times in the manuscript, but I think this just a buzzword that is irrelevant here.
Reviewer 2 Report
I am an avid user and researcher of technology and the use of technology in a different field. Therefore, this research is interesting to me and review. The systematic literature is carried out to demonstrate the factors for adaptation.
However, it could be enhanced with some cases and social evidence to support the systematic review. Therefore, it is important to validate the literature with some evidances.
Author Response
Point: However, it could be enhanced with some cases and social evidence to support the systematic review. Therefore, it is important to validate the literature with some evidances.
Response:
In the “Introduction” section, social evidence is added:
Page2,Line 48: Chung et al. concluded that when smart technology is implemented appropriately and ethically, it can strengthen older adults' quality of life, safety, and prospects for aging-inplace [6]
In the “Influencing factors of the elderly adopting IHMT” section, social evidence is added:
Page5,Line192: A recent survey by AARP(American Association of Retired Person) on technology trends among older adults identified smart technology as an emerging market for the 50+ age group, with many adults interested in buying smart technology products, but only about 10% currently using them[42].

Round 2
Reviewer 1 Report
As mentioned already in my previous review, I want to congratulate the authors for the enormous effort in selecting, reading and extracting info from a large number of relevant journal articles.
The authors have implemented most of my comments in my previous review satisfactorily: the English language has improved greatly (except for a few instances, see below).
My comment about the journal articles are discussed in the text have also been implemented, but not in all cases. I would suggest the authors to go through the text again and change where necessary. Ideally, each of the subchapters should start with a clear summary of their findings, and then support that by a number of referenced articles. And ideally in the form of "Authors-ABC show in their study .... [ref]". Some examples that need to be modified:
> In subchapter 3.1.3 it is stated in lines 172-175: "For patients with chronic illnesses .... we can monitor them (Who is 'we' here???) ..." Change this to: "In their study, Brath et al ..... [45]."
> In subchapter 3.1.4. (line 190 and further): in lines 197-199 an outcome is mentioned from reference [2] is written, but the authors are not mentioned, and it is not clearly stated that this comes from that article. So I would recommend to change this into: "In their study, Zou et al applied a diet... [2]". There are some more examples of this.
A general comment on the overview of types of IMHT (Tabel 1) and Influencing factors (Table 2): in the text of the various subchapters, where all the types and influencing factors are discussed, it is not always easy to see which type and which influencing factor is discussed in that specific part of each subchapter. Therefore I would recommend to underline those keywords, for instance: 'perceived demand' in line 263, and 'self-efficacy' in line 274 (and so forth).
Second general comment (advise) would be to add after the title "1 Demographic characteristics" in line 231 the text "(objective factors)", since that is now the term that is used in Table 2. The same goes for "2. Motivation" (line 261): add "(subjective factors)" here. Or change the complete title.
line 98-100: change the end of the sentence to: ".. and realize that the elderly can be at home safely".
line 109: what is meant with "background' here?
line 136-137: leave out "web-based systems" because that is a much too general term.
line 137-138: "Telemedicine is a type of telehealth" -> What is the difference between telemedicine and telehealth??
line 150-153: An electronic health record system is primarily a system to be used by healthcare providers the register information on the health of patients. As a secondary use, the information can be made available to the patients themselves. Therefore I would suggest to start this sentence as follows" "Electronic health record systems can, apart from being used by healthcare providers, provide personalized services for the elderly.."
line 168-169: Remote monitoring technology can monitor and track (is that the same?) the physical condition of the elderly, not only changes. Information on changes can be transmitted to the remote monitoring center.
Lines 511-516: the two sentences are very hard to understand. Please rewrite them to make these statements better understandable.
Author Response
Point 1: Ideally, each of the subchapters should start with a clear summary of their findings, and then support that by a number of referenced articles. And ideally in the form of "Authors-ABC show in their study .... [ref]". Some examples that need to be modified:
Response:
Each of the subchapters starts with a summary of findings. And the examples are added to the authors’ info. For example, “Authors-Reeder, McManus, Quinn, Willard, Hermanns, et al show in their study that self-monitoring can be realized in the form of medication dispensing devices, embedded sensors, self-management applications, wearable systems, online community care platforms, and physical activity trackers. [8,9,11,15,16,17](lines 101-105)”,
“In Chen, Parker, Taha, Kullgren, McMahon, Portz et al.'s study, medical care mainly provide services in the form of tele-medicine, APP, electronic health record systems, wearable system, physical activity monitor, patient portals [21,23,28,31,32,33]. (lines 144-148)”,
“Authors-Turchioe, Chau, Sheeran, Taha, Creber, Bakas, et al found that the most common are telemonitoring apps, web-based systems, in-home monitors, patient portals, embedded sensors, robotics [35,4,36,38,39,40]. (lines 190-193)”,
“Studies by Zou, Richard, et al demonstrated that an app and a web-based system could carry out health education[2,41]. (lines 220-221)”,
“Authors-Rasche, Taha, Batsis, Irizarry, Stillet, McManus et al. proved that Whether the elderly adopt IHMT was affected by age, education level, region, e-health literacy, information literacy, ethnicity and language, physical difficulties[12,28,48,5,13,9].” (lines 261-263)
“In the studies of Reeder, Quinn, Zhang, Rasche, et al., perceived demand, self-efficacy, compliance, and skeptical attitudes are subjective factors that affect the adoption of IHMT by the elderly[8,11,19,12].” (lines 300-302)
Authors-Kullgren, Zhou, Turchioe, Irizarry, et al. showed that perceived usefulness is reflected in functionality, information quality, personalization, and interactivity [31,22,35,5]. (lines 424-426)
Point 2: > In subchapter 3.1.3 it is stated in lines 172-175: "For patients with chronic illnesses .... we can monitor them (Who is 'we' here???) ..." Change this to: "In their study, Brath et al ..... [45]."
Response:
Change the sentence “For the elderly with chronic illnesses outside of healthcare settings, we can monitor them through the patient-reported outcomes measurement information system.(line 179-181)” To “In their study, Brath et al. found that the patient-reported outcomes measurement information system can monitor the elderly with chronic illnesses outside of healthcare settings.”
Point 3: > In subchapter 3.1.4. (line 190 and further): in lines 197-199 an outcome is mentioned from reference [2] is written, but the authors are not mentioned, and it is not clearly stated that this comes from that article. So I would recommend to change this into: "In their study, Zou et al applied a diet... [2]". There are some more examples of this.
Response:
Change the sentence into: ”In their study, Zou et al. applied a diet education APP that provided proper diet knowledge for elderly users, supported them in forming a healthy diet, and controlling hypertension [2].”
Point 4: A general comment on the overview of types of IMHT (Tabel 1) and Influencing factors (Table 2): in the text of the various subchapters, where all the types and influencing factors are discussed, it is not always easy to see which type and which influencing factor is discussed in that specific part of each subchapter. Therefore I would recommend to underline those keywords, for instance: 'perceived demand' in line 263, and 'self-efficacy' in line 274 (and so forth).
Response:
We are using italics to emphasize important terms. Such as devices for medication management, embedded sensors, self-management APP, online community care platform, etc.
Point 5: Second general comment (advise) would be to add after the title "1 Demographic characteristics" in line 231 the text "(objective factors)", since that is now the term that is used in Table 2. The same goes for "2. Motivation" (line 261): add "(subjective factors)" here. Or change the complete title.
Response :
“1 Demographic characteristics”(line 231) changes title “Objective factors”
"2. Motivation" (line 261) changes title “Subjective factors”
Point 6: line 98-100: change the end of the sentence to: ".and realize that the elderly can be at home safely".
Response:. Change the sentence to: “Self-monitoring is a technology to provide the elderly with the measurement and early warning of various physical health indicators and realize that the elderly can be at home safely.” (line 98-100)
Point 7: line 109: what is meant with "background' here?
Response: Delete the redundant word "background' (line 109). The sentence is following:
“A smartphone APP using embedded sensors can monitor and manage the patient's heart rate, automated discriminate arrhythmia, and atrial fibrillation.”
Point 8: line 136-137: leave out "web-based systems" because that is a much too general term.
Response: Delete the words "web-based systems"(line 136-137)
Point 9: line 137-138: "Telemedicine is a type of telehealth" -> What is the difference between telemedicine and telehealth??
Response: Change the sentence” Telemedicine is a type of telehealth based on advanced information technology. (line 138-139)” to:
“Telemedicine is the remote diagnosis and treatment of the elderly by means of telecommunications technology.”
Point 10: line 150-153: An electronic health record system is primarily a system to be used by healthcare providers the register information on the health of patients. As a secondary use, the information can be made available to the patients themselves. Therefore I would suggest to start this sentence as follows" "Electronic health record systems can, apart from being used by healthcare providers, provide personalized services for the elderly.."
Response: Change the sentence “Electronic health record systems mainly provide personalized services for the elderly through health information exchange, healthcare software, electronic appointment scheduling, etc., finally facilitating their self-efficacy and health management [27,28,30]. (line150-153)” to “Electronic health record systems can, apart from being used by healthcare providers, provide personalized services for the elderly.”
Point 11: line 168-169: Remote monitoring technology can monitor and track (is that the same?) the physical condition of the elderly, not only changes. Information on changes can be transmitted to the remote monitoring center.
Response:
Change the sentence “Remote monitoring technology is capable of monitoring and tracking the physical changes of the elderly through sensors and transmitting the change information to the remote monitoring center, which will be analyzed and given diagnostic opinions by medical staff. ( lines 169-172)” to “Remote monitoring is a technology to enable monitoring of the elderly outside of conventional clinical settings (e.g. in the home), which may increase access to care and decrease healthcare delivery costs.”
Point 12: Lines 511-516: the two sentences are very hard to understand. Please rewrite them to make these statements better understandable.
Response:
Change the sentence “The elderly form an excellent perception of them in the use process, which is necessary to adopt and continue using them. Due to the sensitivity, security, and privacy of data collected and processed by IHMT, it will significantly affect the perception of the elderly and then affect their use intention. ( lines 511-516)” to “Only products that are useful, easy to use, safe, and reliable for the elderly are welcomed by the elderly.”